# Irrelevant auditory and tactile signals, but not visual signals, interact with the target onset and modulate saccade latencies

**Manuel Vidal**[1,2]*, **Andrea Desantis**[3,4], **Laurent Madelain**[1,5]

**1** Institut de Neurosciences de la Timone, UMR 7289, CNRS, Aix-Marseille Université, Marseille, France,
**2** Laboratoire de Psychologie Cognitive, UMR 7290, CNRS, Aix-Marseille Université, Marseille, France,
**3** Département Traitement de l'Information et Systèmes, ONERA, Salon-de-Provence, France, **4** Integrative Neuroscience and Cognition Center, UMR 8002, CNRS, Université Paris Descartes, Paris, France,
**5** Sciences Cognitives et Sciences Affectives, UMR 9193, CNRS, Université de Lille, Lille, France

* manuel.vidal@univ-amu.fr

**Data Availability Statement:** All relevant data are within the paper and its Supporting Information files. However the raw data files, global plots, individual plots and data processing code for all experiments can be found in the following public

## Abstract

Saccadic eye movements bring events of interest to the center of the retina, enabling detailed visual analysis. This study explored whether irrelevant auditory (experiments A, B & F), visual (C & D) or tactile signals (E & F) delivered around the onset of a visual target modulates saccade latency. Participants were instructed to execute a quick saccade toward a target stepping left or right from a fixation position. We observed an interaction between auditory beeps or tactile vibrations and the oculomotor reaction that included two components: a warning effect resulting in faster saccades when the signal and the target were presented simultaneously; and a modulation effect with shorter–or longer–latencies when auditory and tactile signals were delivered before–or after–the target onset. Combining both modalities only increased the modulation effect to a limited extent, pointing to a saturation of the multisensory interaction with the motor control. Interestingly, irrelevant visual stimuli (black background or isoluminant noise strips in peripheral vision, flashed for 10 ms) increased saccade latency whether they were presented just before or after target onset. The lack of latency reduction with visual signals suggests that the modulation observed in the auditory and tactile experiments was not related to priming effects but rather to low-level audio- and tactile-visual integration. The increase in saccade latency observed with irrelevant visual stimuli is discussed in relation to saccadic inhibition. Our results demonstrate that signals conveying no information regarding where and when a visual target would appear modulate saccadic reactivity, much like in multisensory temporal binding, but only when these signals come from a different modality.

## Introduction

Our environment continuously provides information through physical signals that are transduced and processed by various sensory systems. Although the study of human perception has long focused on isolated senses, in the last decades the interaction between different sensory

repository: https://amubox.univ-amu.fr/s/PFHTMzb6i7emqfc.

**Funding:** This research was in part funded by the Agence Nationale de la Recherche (grant number ANR-18-CE10-0001) awarded to Andrea Desantis. The funders had no role in study design, data collection and analysis, decision to publish, or preparation of the manuscript.

**Competing interests:** The authors have declared that no competing interests exist.

systems has started to gain interest. Many scientists now believe that the nature of purely visual contexts studied in the laboratory is quite different from the multisensory scenarios found in more ecological settings; and that vision per se is often influenced at early stages by other modalities such as audition. Let us consider a daily life situation: as we wander in a fun fair the sudden explosion of a balloon on the right side of our visual field will automatically attract our gaze. This orientation behavior facilitates further sensory processing needed for a fast reaction to a potential danger. The resulting brief acoustic signal and salient visual change generated by the explosion are transmitted in the air and then transduced by our auditory and visual systems at different speeds. Although these signals reach the brain areas responsible for their integration at different moments [1,2], we will most likely perceive them as simultaneous. Moreover, depending on the distance to the blowing balloon, the auditory signal might reach the observer either before or after the visual signal. The mechanisms involved in the subjective timing of natural events must then show some degree of flexibility to determine whether each unimodal component originate from the same causal source, before combining them into a single multisensory event (for a detailed review on the causality in perception, refer to [3]). Furthermore, the observed advantage of the visual modality over other senses for spatial processing does not hold for the temporal domain: the auditory system is more sensitive and reliable than the visual system to process the timing of events [4]. When presenting a flash just before or after a short beep, the visual stimulus is perceived closer in time to the auditory stimulus than it actually is [5]. Consistently, the perception of a flash shifts either forward or backward in time when paired with a lagging or leading sound click, respectively [6]. This perceptual phenomenon could have consequences in visually-guided motor tasks. This is nicely illustrated by the spatial shift of a visual flash that is presented after a saccade in conjunction with a tone [7]. The flash is mislocalized as if it had occurred closer in time to the tone. However, a recent study showed that in a visually-guided hand motion task, dynamic motion estimation relies almost exclusively on faster proprioceptive feedback rather than on the more accurate visual feedback [8]. This puzzling finding shows that the brain considers both sensory variance and temporal delays to compute the optimal motor behavior. Back to our example, the blowing sound would affects the moment when we visually perceive the visual counterpart of the balloon explosion, the latter being shifted in time toward the auditory signal. This could have consequences on the timing of the saccade that would be programmed to bring the retinal image of the event onto the fovea. For instance, if the balloon is distant enough, the blowing sound will reach our ears and be processed after the visual signal. The temporal multisensory binding will then delay the moment we perceive the explosion. Would this delay the execution of eye movements towards the location of the explosion as well? In other words, does this temporal binding also influence the motor response latency executed in reaction to the multisensory event, or will the motor response be mostly visually guided, regardless of the audiovisual perceptual illusion as suggested by [8]?

We addressed this questions in a series of experiments using saccadic eye movements as a probe model, for they allow measuring small differences in processing time [9]. In addition, these eye movements are partly controlled by the superior colliculus (SC), a well-known brain structure also involved in multisensory processing. In mammals, the SC in which sensory and motor maps are connected transforms sensory inputs into motor commands [10–12]. The mechanism underlying gaze fixation has been proposed to involve fixation cells in the rostral SC, which inhibit the generation of saccades through the excitation of omnipause neurons in the brainstem [13]. Importantly, SC was also the first structure where audiovisual integration in time and space was observed [14–16]: neurons in the superficial SC layers respond to visual stimuli and neurons in the intermediate and deep SC layers also respond to auditory stimuli, such that auditory and visual sensory maps are connected at a very early processing stage.

Audiovisual interactions in monkey SC modulate saccade-related activity, though less than what expected from earlier recordings [17]. This physiological organization has functional consequences that might be observed in behavioral studies.

Only a handful of studies have investigated the influence of signals from non-visual sensory modalities on the execution of saccades. Ross & Ross [18] used continuous warning signals–onset or offset of either sounds or visual symbols at fixation–around the time of target onset. Saccade latencies were shorter for auditory signals presented 300 ms, 100 ms or 0 ms before target onset, indistinctly for sound onset or offset, when compared to latencies in conditions without warning signals. However, the nature of these effects proved to be different from what was observed with visual signals as saccades were delayed when visual stimuli were displayed after target onset. Moreover, onset and offset of visual signals differentially affected the execution of saccades. However, the study used a group design and lacked data points precisely where strongest multisensory interactions are expected to happen, that is, for stimulus onset asynchronies (SOA) within [−100 ms; +100 ms]). Another study reported a facilitation modulated with SOA ranging between −50 ms and +100 ms for spatially congruent sounds, which disappears with incongruent sounds [19]. However, the limited number of participants (N = 3) constrained statistical conclusions and weakened the impact of the study. Similar multisensory latency variations have been reported for SOAs ranging from −30 ms to +120 ms, again modulated by the spatial congruency between the sound and the visual target [20]. These multisensory influences have an effect on saccades that could share similar mechanisms and combine with the gap/overlap effect [21]. Saccade latencies decrease when fixation disappears before target onset (gap) and increase when fixation remains after target onset (overlap).

Another line of research focused on the effects of spatial congruency across modalities. Saccades toward a visual target had shorter or longer latencies when an auditory distractor was spatially aligned or misaligned, respectively [22]. These effects depend on the physical distance between the visual target and the sound source [19,20]. For combined congruent audiovisual targets, the latency decrease was well predicted by independent race models, suggesting that multisensory integration in the target selection is not optimal [22]. Finally, inhibiting saccades toward an auditory distractor when the fixation point is already turned off proved more difficult than when the fixation remains visible at the target onset [23], revealing an interaction between disengaging fixation and the target selection in the SC. Importantly, tactile stimulations as well influence saccades, both in the spatial and temporal dimensions [24]: latencies are reduced when a touch is delivered before target onset, the facilitation being maximal when the touch and visual target are spatially congruent [25]. Reactive saccades (i.e. stimulus-driven) have been observed using purely somatosensory stimuli [26], which points to early interactions of the visual and tactile modalities in the generation of saccades. The best locus candidate being again the deep layers of SC [27,28].

The current project brings new insights into the multisensory interactions occurring when programming and executing eye-movements. We investigated whether an irrelevant stimulus delivered around the appearance of a visual target, which expectedly alters the perceived timing of the target onset, might influence saccade latencies. Specifically, we used auditory (experiments A and B), visual (experiments C and D), tactile (experiment E) and combined tactile and auditory stimuli (experiment F) to probe the effects of multisensory temporal integration on the possible mechanisms underlying the selection and execution of saccades. Our results indicate reliable SOA dependent multisensory effects on saccadic reaction time when using sounds or touches. Visual stimuli, on the other hand, appear to produce a different behavioral pattern, indicating that temporal integration requires signals from different modalities.

## Experiment A–Beeps & Saccades

The goal of the first experiment was to test whether a short beep presented in temporal proximity with the onset of the visual target alters the execution of saccades. We measured the latencies of saccades toward visual targets appearing either rightward or leftward of a fixation, while delivering a beep with a stimulus onset asynchrony (SOA) ranging from −240 ms (beep first) to +240 ms (beep after). Saccadic eye-movements were chosen for they provide fast and reliable responses that allow measuring small differences in decision processing delays. A baseline condition without beep was also tested.

## Materials and methods

**Ethics statement.**   For each experiment of this project, subjects gave a prior written consent after being informed of the methods used and their right to interrupt if they wished. This project was approved by the Comité d'éthique d'Aix-Marseille Université (reference 2014-12-3-06) and complies with the regulations described in the Declaration of Helsinki (2012).

**Participants.**   8 volunteers (5 women and 3 men) participated in this experiment, all naïve to the purpose of the experiment except two of the authors. Participants were aged between 22 and 43 years old (average 32.3). They all had normal or corrected-to-normal visual acuity and were all right handed.

**Apparatus and stimuli.**   Subjects sat in front of a screen with head movements restricted by a chin and head rest. Stimuli were generated on a Mac computer running Mac OS 10.6.8 operating system. Routines were written in Matlab 7.10.0 using the PsychToolbox 3.0.9 [29,30]. The right eye position was recorded using an SR Research EyeLink 1000 video eye tracker (sampling at 1000Hz) mounted on the same structure as the chin rest. Visual stimuli were displayed on a Sony Trinitron CRT monitor running at a resolution of 1024×768 and refreshed at 100Hz (frames of 10 ms). The chin rest was adjusted so that the eyes in primary position were aligned with the center of the screen, at a distance of 57 cm. The fixation point was a small white disk (0.12˚ in diameter) displayed at the center of the screen. The saccade target was a white disk (0.36˚ in diameter) that could appear either to the left or to the right of the fixation at an eccentricity of 8˚. Background was set to 50% grey level (25.8 cd/m$^2$ luminance after gamma correction). Beeps were 20 ms 880 Hz tones attenuated by a raised-cosine waveform (50% after 10 ms) delivered binaurally through closed headphones (Beyerdynamics DT770). The computer audio driver was set so that the audiovisual jitter remained below 1 ms. The accuracy of the timing of visual and auditory stimuli was controlled using a dual-channel oscilloscope connected to both the auditory output and a photosensitive cell placed directly on the screen.

**Procedure.**   Fig 1 illustrates the general time course of a trial with the various signals used for each of the six experiments we conducted. Trials started with the fixation point appearing in the center of the screen. Subjects were asked to fixate it and to avoid blinking during the stimulus presentation. After a random delay ranging from 750 ms to 1250 ms, the fixation was turned off and the saccade target appeared either leftward or rightward, at an eccentricity of 8˚. In most trials of experiment A a beep was delivered around the time of target onset. Trial conditions were defined as the combination of two factors: 10 values of stimulus onset asynchronies (SOA) i.e. the delay separating the onset of the visual target and the auditory stimulus (−240, −120, −60, −30, 0, +30, +60, +120, +240 ms and *No beep* baseline where no beep was played) and 2 target directions (left and right). Subjects were instructed to perform a saccade as quickly and accurately as possible toward the visual target. They were explicitly told that beeps were irrelevant to the task and not to pay attention to them.

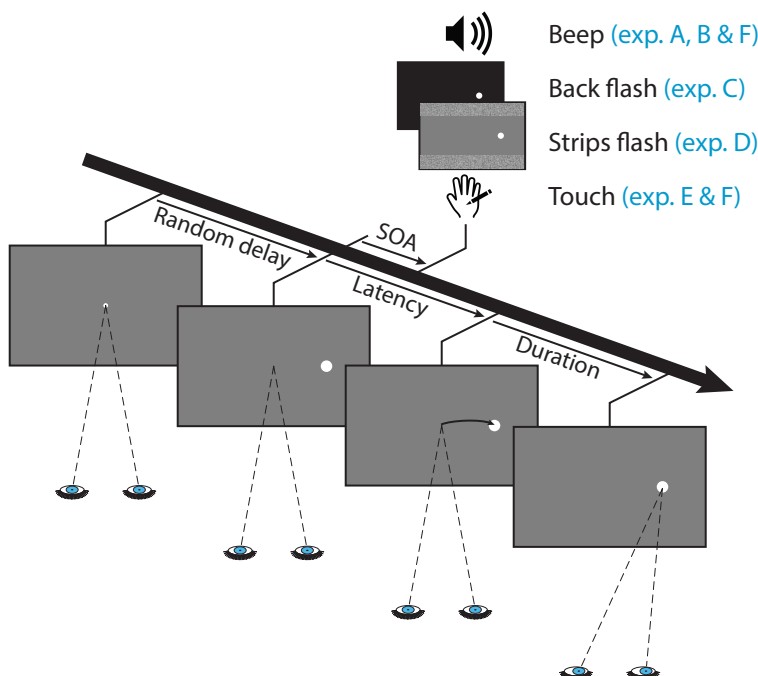

**Fig 1. Time course of a trial with the different signals used for each experiment.** A central fixation point is presented for a random duration (ranging from 750 to 1250 ms) then the target appears either to the left or to the right and a brief signal is delivered either before or after target onset (SOA ranging from −240 to +240 ms). The signal was either a beep (exp. A, B & F), a background luminance decrease (exp. C), a pair of isoluminant strips (exp. D) or a tactile vibration (exp. E & F), all lasting about 10 ms. In trials with the baseline conditions, no signal was delivered. The participant had to shift gaze toward the target as fast as possible, regardless of the non-informative signal. The saccade latency is defined as the delay between the target appearance and the eye-movement onset; the saccade duration is the time needed to land on the target.

The first five subjects completed 8 sessions of 300 trials totaling 2100 trials (105 per SOA × direction). For the remaining three subjects, the experiment was reduced to 4 sessions of 240 trials totaling 960 trials (48 trials per SOA × direction). The order of the conditions was randomized within blocks of 40 trials. Between each session subjects had a few minutes break where they could stand out of the setup to rest. The calibration procedure (using 13 positions on the monitor) was performed at the beginning of each session. After each block of 50 trials, subjects could rest for a few seconds without moving their heads, then a potential drift in eye movement calibration was checked (using a single location) and corrected if needed, before resuming.

**Data processing.** We used the Eyelink online saccade detector to identify saccades onset and offset, using 30°/s velocity and 8000°/s$^2$ acceleration thresholds [31]. Invalid trials in which either no saccade was detected, eyes blinked, saccades started too late (latency>400 ms) or fell too short (amplitude<3°)—were discarded. An adaptive low-pass filter was then applied to the set of latencies in order to remove most of anticipatory saccades. For each condition, the cutoff criterion of the latency distribution was determined using an iterative process. The cutoff started at 80 ms and increased by steps of 1 ms until fewer than 1% of the remaining saccades in the distribution were not directed toward the target. Because anticipatory saccades have 50% chances to go in the wrong direction, limiting these saccades below 1% mechanically limited anticipatory saccades going in the right direction below 1% as well (see the discussion below for more details on this issue). In this experiment, the cutoff was set at the lowest value

(80 ms), with the proportions of saccades in the wrong direction averaging 0.10% across participants (maximum 0.29%). This filter was designed to adapt to individual peculiarities and find the optimal tradeoff between a maximum of visually-driven saccades in the distributions while removing a maximum of saccades programmed before processing the visual signal related to target onset. At this stage, only two saccades going in the wrong direction were detected across all subjects. Finally, saccades falling short (gain<0.45) and going in the wrong direction were excluded. A total of 12327 saccades out of the 13080 recorded trials were analyzed (94.2%). For each subject and each SOA condition, the median value and the median absolute deviation (MAD) of the saccade latency distribution were computed. In order to reduce inter-individual dispersion for the statistical analyses, and to allow for comparisons between experiments, we have normalized the median latency and the MAD taking the *No* signal condition (i.e. *No beep* in experiment A) as a baseline, using the following equations:

$$n_{score}(subject, SOA) = \frac{x(SOA) - x(No\ signal)}{\sigma_x(subject)} \quad (1)$$

where $x$ is either the median latency or the MAD of a given condition, and $\sigma_x$ is the standard deviation of $x$ across all conditions. A two-way repeated measure ANOVA (design: 9 SOA × 2 target direction) was performed on the $n_{scores}$. The *No beep* condition, that always had a null $n_{score}$ by construction, was excluded from the design. There was neither a main effect of Target direction ($F(1,7) = 2.17$, $p = 0.18$) nor a SOA x Target direction interaction ($F(8,56) = 0.77$, $p = 0.63$). Consequently, left and right saccades were pooled together in a single distribution and the median of this collapsed distribution was further analyzed using a one-way repeated measure ANOVA with SOA as factor. Planned comparisons consisted of paired student t-tests between SOA conditions, and a single sample t-test to test the difference with the *No beep* baseline, as detailed below. To ensure that $n_{scores}$ did not deviate from normality and that they met the homoscedasticity assumption required for parametric statistics, we conducted a Shapiro-Wilk's test and a Levene's test before performing these planned comparisons. Except otherwise stated, throughout this article all sets of measures had distributions that did not significantly differ from normality, and the compared sets had homogeneous variances. The alpha value for significance was set to 0.00555 after Bonferroni correction for multiple comparisons on a single data set (here 9 comparisons). The results of all these tests along with their corresponding effect sizes (Cohen's d) are reported in **S1 Table** (experiments A to E) and in **S2 Table** (experiment F). Raw data files, global plots, individual plots and data processing code for all experiments can be found in the following public repository:

*https://amubox.univ-amu.fr/s/PFHTMzb6i7emqfc*

## Results

The effects of auditory beeps on the saccade onset as a function of SOA are summarized in **Fig 2**. The general pattern of individual median latencies–plotted in grey–show substantial similarities across subjects, with some inter-individual differences in the amplitude of the effects. To reduce this disparity we computed, for each SOA condition, the $n_{scores}$ taking the *No beep* condition as a baseline, which revealed a high degree of homogeneity in participants' behaviors. SOA had a significant main effect on the $n_{scores}$ ($F(8,56) = 55.50$, $p<0.0001$, $\eta_p^2 = 0.89$). In order to evaluate the extent of the warning effect when a beep is delivered near target onset, we performed single sample t-tests with the $n_{scores}$ observed in the SOA = 0 ms condition and with the average $n_{scores}$ across the range of SOAs. This analysis revealed that saccade latency decreased significantly by about 11.0 ms when the target and the beep were presented simultaneously ($p<0.0002$) and by 7.6 ms on average (p<0.00001), compared to the *No beep*

**Fig 2. Experiment A. Effect of sound beeps on saccade onset.** Median latencies **(left panel)** and median absolute deviations **(right panel)** averaged across participants for each SOA condition (top) and their corresponding $n_{scores}$ (bottom). Dashed lines show the *No beep* condition level and grey lines show individual results. Error bars indicate inter-individual SEM. Statistics included a single sample t-test performed on the $n_{scores}$ of the SOA = 0 ms reference condition to highlight the difference with the *No beep* condition (black arrow), and paired t-tests comparing this reference with each of the other SOA condition (red stars for each SOA above the X-axis). Three stars indicate highly significant differences after Bonferroni correction (p<0.00555) while single stars significant differences without correction (p<0.05). Red arrows indicate the overall range of variations.

condition. To further investigate possible modulations of saccade latency we then performed paired t-tests on the $n_{scores}$ comparing the SOA = 0 ms reference with each of the other SOA condition (see **S1 Table** for a detailed report of these tests). Saccade latencies tended to be shorter for earlier beeps (i.e. beep onset precedes target onset) compared to synchronous beeps (*p* = 0.024 and *p* = 0.038 for SOA = −120 and −60 ms respectively) and increased significantly for later beeps (i.e. beep onset follows target onset) (*p*<0.0001 for all positive SOA). This modulation produced saccades that tended to have even longer latencies than the *No beep* baseline for SOA = +60 ms and +120 ms (paired t-tests, *p* = 0.011 and *p* = 0.008), before returning to the asymptotic level of the baseline at SOA = +240 ms. Within the studied range of SOA, beeps modulated the saccadic reaction time by about 22 ms, corresponding to 16% of the median latency observed with synchronous auditory and visual stimuli.

Finally, we performed the same analyses on the median absolute deviation (right panel) to determine the effect of the beeps on the temporal precision of saccades. Interestingly, delivering a synchronous beep significantly reduced latency dispersion (p<0.003) compared to the *No beep* baseline, with a dispersion for SOA = 0 ms significantly smaller than for SOA = −120 ms (p<0.004), +60 ms (p<0.002) or +120 ms (p<0.0004).

## Discussion

Saccades were triggered sooner by about 11ms when a spatially non-informative beep was delivered precisely at the target onset (SOA = 0 ms). This reduction in reaction time could be the signature of increased attention: the external sound would act as a warning event that speeds-up saccade triggering [18,32]. Another interpretation might propose that the sound acts as a distractor that partially disengages the attention allocated to the fixation point, thereby reducing latencies with mechanisms similar to those possibly involved in a gap paradigm [33]. Moreover, early beeps tended to reduce the saccade latencies compared to the synchronous beep condition, while late beeps significantly delayed the saccades for all positive SOAs, producing saccades with even longer latencies than when no beep was delivered. For beeps occurring 240 ms after target onset, saccade latencies decreased back to the baseline asymptote. For such large audiovisual discrepancy, beep and target are not bound together [4] indicating a break in the causal link between the two events [3].

It is noteworthy that subjects sometimes initiated their saccades before having correctly processed where the target appeared, mostly in conditions where the beep was presented before target onset. This could possibly reduce the latencies for negative SOAs and produce the overall pattern we observed. However, we believe that this was not the case. Saccades triggered before proper target detection have 50% chance of going in the right direction. To remove these anticipatory saccades, we used a similar reasoning as what has been proposed to isolate true express saccades [33]. Individual histograms showing, for each condition, the number of initial saccades going in the opposite direction–plotted either before or after applying the adaptive low-pass filter on latencies described in the Materials and Methods–are provided in **S1 Fig** of the supporting information. Only 12 saccades going in the wrong direction remained, a negligible fraction of the distribution (considering the 12327 that were analyzed). Therefore, we are confident that the vast majority of saccades programmed before target detection (but going in the right direction) were also removed by our low-pass filter. Moreover, unlike for means, the analysis of medians is rather robust to extreme values. We can therefore conclude that anticipatory saccades did not significantly contribute to the observed latency modulations.

In this experiment, we presented a sound in the majority of trials and participants might have developed an expectation such that in each trial they waited for a sound before triggering the saccade. This strategic behavior would obviously lead to increased latencies in trials in which the sound was played after the visual target or not played at all. One could therefore argue that the effect we observed might be simply induced by the experimental design. In order rule out this possibility, we conducted a control experiment in which the experimental conditions were blocked to better control the participant's expectation.

## Experiment B–Blocked control

This experiment used the same method as experiment A except that conditions now were blocked and only 3 SOAs (−60, 0 and +60 ms) and the *No beep* condition were used. In other words each SOA condition and the *No beep* condition were presented in separate blocks. If the warning effect found in experiment A was due to delayed saccades in the *No beep* condition rather than shorter latencies with synchronous beeps, this advantage should be reduced or cancelled when blocking the experimental conditions. Similarly, if the modulation effect found in experiment A was due to waiting for the late arrival of beeps, one would expect to find a limited difference between the SOA = 0 ms and the other positive SOA conditions with a blocked design. 6 volunteers (2 women and 4 men) participated in this experiment, 4 of them participated in experiment A. They were aged between 24 and 43 years old (average 35.8), all had

normal or corrected-to-normal visual acuity and all but one were right handed. Subjects completed 2 sessions of 240 trials totalizing 480 trials (48 per SOA × direction). Each block included 60 trials with a single SOA and two possible target directions. The order of these 60-trials blocks was randomized for each participant.

## Results

A total of 2727 out of 2880 recorded trials were analyzed (94.7%). After applying our low-pass filter to remove anticipatory saccades, only 3 saccades going in the wrong direction remained. Given that the latencies for leftward and rightward saccades did not differ–no main effect ($F$ $(1,5) = 0.25$, $p = 0.88$) nor interaction with SOA ($F(2,10) = 1.82$, $p = 0.21$)–we pooled them together for further analyses. Latency $n_{scores}$ were computed for each of the 3 SOA conditions taking the *No beep* condition as a baseline.

Fig 3 summarizes the results of experiment B, showing the effect of beeps on saccade latency when the SOA conditions were blocked (left panel) and comparing it with the results of experiment A where conditions were interleaved (right panel). In experiment B, SOA had again a significant main effect on the $n_{scores}$ ($F(2,10) = 154.44$, $p<0.0001$, $\eta_p^2 = 0.97$). In order to evaluate whether saccade latency was affected by the beeps, we performed single sample t-tests on the $n_{scores}$ of the SOA = 0 ms condition and on the average $n_{scores}$ across the range of SOAs. The analysis showed that saccade latency significantly decreased by about 13.5 ms with a synchronous beep ($p<0.001$) and by 8.9 ms on average (p = 0.018), compared to the no beep condition. We then compared latencies from the SOA = 0 ms reference condition with the other two SOAs using Student's paired t-tests (see S1 Table for a detailed report of these tests). The analyses showed that early beeps tended to have shorter latencies ($p = 0.07$ for SOA = −60 ms), while late beeps significantly increased saccade latencies ($p<0.0001$ for SOA = +60 ms). In order to compare data obtained in experiment B with the ones from experiment A, we computed for both experiments the individual median latency differences between each SOA condition and its respective *No beep* baseline. Unpaired Student t-tests comparing the latency differences obtained for each experiment revealed no significant difference (see S3 Table). Moreover, the average modulation effect within the same range (i.e., −60 to +60 ms) was not different (19.6 ms vs. 18.8 ms for experiment B and A, respectively).

## Discussion

Grouping the various trial types in blocks should promote forming expectations about when exactly the beep would appear with respect to target onset. We did not observe a reduction of the advantage found in experiment A when delivering a synchronous beep, nor limited difference between the SOA = 0 ms and the other positive SOA conditions. The absence of differences in the blocked versus interleaved experiments provides a clear indication that the modulation observed in experiment A is not a design artifact related to a strategic waiting for the auditory signal. Overall, these results support the view that the effects on saccade latency result from multisensory interaction influencing the temporal processing of the target. Auditory and visual maps encoding saccadic goals in space have been found in the deep layers of superior colliculus [10,11]. These maps are spatially kept in register permitting efficient audio-visual integration [34]. The evidence for integration in the temporal dimension reported here could take place in the superior colliculus as well. Indeed, auditory events delivered near the timing of target onset would produce a stronger multisensory activity in the motor maps coding for the target in SC [19].

Interestingly, multisensory interactions influenced the temporal processing of the target, even though the beeps did not provide any information regarding the direction or timing of

**Fig 3. Experiment B. Effect of sound beeps on saccade onset with blocked conditions (left panel).** Median latencies averaged across participants for each SOA condition (top) and the corresponding n_scores (bottom). Dashed lines show the *No beep* condition level and grey lines show individual data. Error bars indicate inter-individual SEM. Statistics included a single sample t-test performed on the n_scores of the SOA = 0 ms reference condition to highlight the difference with the *No beep* condition (black arrow), and two paired t-tests comparing this reference with SOA = −60 or +60 ms conditions (red stars above the X-axis). **Interleaved vs. blocked conditions (right panel).** Comparison between experiment B (red plot) and A (grey plot): median latency differences (top) and n_score differences (bottom) aligned on the *No beep* condition (dashed lines). Unpaired t-tests performed on the n_score differences between experiments for the 3 SOA conditions showed no significant difference. Three stars indicate highly significant differences after Bonferroni correction (p<0.01666). Red arrows indicate the overall range of variations.

the target. In an earlier study, we found that the temporal integration of a beep and a flash can modulate the perceived appearance of the flash, so that it is perceived later or, to some extent, earlier in time, depending on whether it is preceded or followed by a beep [4]. Such asymmetrical modulation was also reported for multisensory integration after adaptation to an audiovisual temporal lag [35–37]. Here we found a similar asymmetrical modulations of saccade triggering, suggesting that the same multisensory mechanisms might be involved. Importantly, two interaction modes depending on the audiovisual offset are possible [4]: for offsets below 40 ms, a single fused bimodal event is perceived while for larger offsets and up to 200 ms, two separate events are perceived but they are reported as being closer in time than they physically are. Supposing that the same perceptual law applies to the target onset, we propose that for short SOAs the beep and the target were perceived as being simultaneous while for longer SOAs the perceived target is only shifted in time toward the beep. Consequently, participants might have detected the visual target faster or slower depending on whether the auditory signal preceded or followed the target, thus modulating saccade latency. It should however be pointed out that the extent of latencies modulations reported here (about 12 ms) is much

smaller than what has been found in purely perceptual tasks [about 120 ms, 4], a result that will be further discussed in the general discussion.

## Experiment C–Background flashes & Saccades

To probe whether the modulation effects we found in the two previous experiments is specific to audiovisual integration and to disentangle it from possible alertness effects, we conducted another experiment in which the auditory signal (beep) was replaced by a visual signal. The visual signal was a full-screen luminosity decrease from the usual grey background (25.8 cd/$m^2$) to a totally black background (0 cd/$m^2$) lasting one frame (10 ms) before returning to the original grey value. We used a transient signal–as for the previously used auditory signal–and, since we operate in the visual modality, we opted for a signal that carries no spatial information to prevent possible interference with saccadic control mechanisms. Indeed, we reasoned that in the studies of Ross & Ross [18,38] the onset and offset of symbols might compete with the encoding of the target location, possibly taking place in the superior colliculus, resulting in specific interactions not related to a multisensory combination. The experimental protocol and data processing were otherwise the same as in experiment A. 8 volunteers (5 women and 3 men) participated in this experiment, 7 of which already participated in experiment A. They were between 25 and 43 years old (average 31.4), all had normal or corrected-to-normal visual acuity and all were right handed.

### Results

A total of 7354 out of 7680 recorded trials were analyzed (95.8%). After applying our adaptive low-pass filter only 6 saccades going in the wrong direction were not removed, showing that the vast majority of anticipatory saccades were successfully eliminated. Since no difference was observed in the latencies of leftward versus rightward saccades–no main effect ($F(1,7) = 0.041$, $p = 0.84$) nor interaction with SOA ($F(8,56) = 1.17$, $p = 0.33$)–they were pooled together for further analyses. Latency $n_{scores}$ were computed for each of the 9 SOA conditions taking the *No flash* condition as a baseline. **Fig 4** summarizes the results of experiment C, showing the effect of background flashes on saccadic onset (left panel) and comparing it with the effect of beeps reported in experiment A (right panel). SOA had a significant main effect on the $n_{scores}$ ($F(8,56) = 23.84$, $p<0.0001$, $\eta_p^2 = 0.77$). Single sample t-tests on the SOA = 0 ms condition $n_{scores}$ and on the average $n_{scores}$ across the SOA range showed that neither synchronous flashes ($p = 0.52$) nor flashes in general ($p = 0.34$) affected saccade latencies, compared to the *No flash* condition. Moreover, student t-tests comparing the SOA = 0 ms reference with each of the other SOA condition (see **S1 Table**) showed that flashes increased saccade latency both when they were presented just before ($p = 0.020$ for SOA = −30 ms) and after target onset ($p<0.003$ and $p<0.0004$ for SOA = +30 and +60 ms, respectively). This reactivity impairment is confirmed by the reduced accuracy observed in the saccade landing positions: saccadic gains were significantly lower when flashes were delivered 30 ms before or 120 ms after target onset ($p<0.004$, see supporting information **S2 Fig** left). In order to compare the results from experiment C with the ones from experiment A, we computed the individual median latency differences between each SOA condition and the *No beep* (experiment A) or *No flash* (experiment C) baselines. We then performed unpaired Student t-tests to compare these latency differences between experiments (see **S3 Table**). We found a qualitatively different pattern, with nearly all SOA conditions yielding significantly higher latencies in experiment C compared to experiment A ($p<0.0025$ for all except SOA = −60, +120 and +240 ms with $p = 0.0070$, $p = 0.036$ and $p = 0.68$ respectively). The maximum average difference across the SOA range was only marginally different (31.3 ms vs. 22.2 ms, $p = 0.063$ computed on the $n_{scores}$).

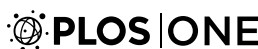

**Fig 4. Experiment C. Effect of background flashes on saccade onset (left panel).** Median latencies averaged across participants for each SOA condition (top) and the corresponding $n_{scores}$ (bottom). Dashed lines show the *No flash* condition level and grey lines show individual results. Error bars indicate inter-individual SEM. Statistics included a single sample t-test performed on the $n_{scores}$ of the SOA = 0 ms reference condition to highlight the difference with the *No flash* condition (black arrow), and paired t-tests comparing this reference with each of the other SOA condition (red stars for each SOA above the X-axis). **Auditory beeps vs. visual background flashes (right panel).** Comparison between experiment C (red plot) and A (grey plot): median latency differences (top) and $n_{score}$ differences (bottom) aligned on the *No signal* conditions (dashed lines). Statistics were unpaired t-tests performed on the $n_{score}$ differences between experiments for each of the 9 SOA conditions. Three stars indicate highly significant differences after Bonferroni correction (p<0.00555), single stars and two stars significant differences without correction (p<0.05 and p<0.01 respectively). Red arrows indicate the overall range of variations.

## Discussion

In this experiment, the data show that when a visual signal–a background flash–was presented 30 to 60 ms after target onset, saccades were strongly impaired with longer latencies and lower spatial accuracies. When the background flash was presented 30 ms before target onset, saccades were also impaired albeit to a limited extent. However, when the visual signal and the target onset were simultaneous we observed similar saccade latencies both when the target was presented alone and when it was combined with a simultaneous flash. However, when the visual signal occurred between 60 to 120 ms before target onset, saccade latency tended to decrease. These findings could be regarded as related to the effects induced by visual warning signals–onset, offset or change of symbol 'o' within the fixation cross–on saccades [18,38]. The authors reported a warning effect when the visual signal was delivered 100 ms to 600 ms before target onset, with shorter latencies than without a visual signal. Conversely, when the visual signal was delivered 50 to 150 ms after target onset, saccade execution was impaired and

latencies were longer. Furthermore, they found differential effects between onset and offset signals on saccade latencies, which points to underlying mechanisms similar to the ones involved in the gap and overlap effects [21]. Indeed, in the Ross and Ross [18,38] paradigm, visual signals inside the fixation cross that appeared before (or disappeared after) target onset could have impaired (or facilitated) the release of fixation by modulating the activity of the rostral pole of the superior colliculus responsible for fixational eye movements [39]. To avoid such interactions between visual signal, fixation and target, we used a full-background flash conveying no spatial information. Despite this important methodological difference we also found longer saccade latencies when the flash was presented 30 ms before and 30 or 60 ms after target onset. However, we found a limited warning effect when the flash appeared more than 60 ms before target onset; and no changed in latencies when the flash coincided with the target onset. Curiously, White et al. [40] found a strong warning effect using a large pink noise visual distractor presented before target onset. In their study, the visual signal remained on until the end of the trial, which might explain why they did not observe the same detrimental interference we did. Indeed, the rapid transient feature of our visual signal probably comes with a stronger propensity to divert attention. Moreover, in their study only the fixation-to-target SOA was manipulated, inducing a gap or overlap situation, while the distractor always appeared at the offset of fixation. Therefore, the effect of the visual signal timing relative to the target onset–our SOA–could only be determined indirectly by subtracting the gap/overlap effect obtained in baseline conditions, reducing the validity of this result.

Because the transient visual signal we used did not provide a fixation signal that could compete with the target, the changes in saccade latencies we observe here cannot result from a difficulty to release fixation, as in it is thought to be the case in an overlap paradigm. Instead, in our experiment the reduced saccade performance could relate to masking effects produced by saccadic inhibition. On the one hand, it is well established that when a visual distractor is presented simultaneously with the target, saccade latencies increase depending on the distractor's position in the visual field [41,42], a phenomenon called the remote distractor effect (RDE). On the other hand, a large peripheral distractor presented shortly after the target strongly reduces the probability of triggering a saccade around 100 ms after the distractor appearance, a phenomenon called saccadic inhibition (SI) which results in a dip in the saccade latency distributions [43]. It has been proposed that these are one and the same phenomenon, where SI would be the underlying mechanism of the RDE [44,45]. In our experiment, when adding a full screen visual flash shortly after the target onset we create the condition where SI takes place. Indeed, we found delayed saccades with latencies peaking when the flash occurred about 60 ms after target onset. This is consistent with the SI reported for transient large top/bottom flashes [43], persistent large arrays of targets [46], but not persistent large pink noise stimuli [40] where backward masking, once isolated from the gap/overlap baseline, was only very limited. As discussed earlier, this difference could relate to the transient nature of our signal, but also to different choices in the design, which allowed us to assess directly the effect of the signal offset on latencies. Lastly, the reduced gain of saccade amplitudes observed when the flash was presented about 100 ms after the target onset is yet another signature of SI [46]. Interestingly, we found no saccadic impairment when both flash and target onset occurred simultaneously, which could be interpreted as the absence of RDE when using large distractors [40]. However, in such situation, the drastic change in luminance contrast when the background screen turns black increases the saliency of the target, which might reduce its processing time: the RDE could be compensated by an opposite effect due to the increased visibility of the target.

Overall, we found a limited visual warning effect: adding background flashes before the target onset did not lead to shorter saccadic latencies, except for the 120 ms SOA. We could not find evidence for visual-visual temporal integration either: irrelevant background flashes did

not attract the target onset in time to modulate saccade latencies in the same way as beeps did. As we proposed, the latency modulation observed with auditory signals could result from audiovisual integration, taking place most probably in the deep layers of the superior colliculus. We now suggest that mechanisms other than multisensory integration are involved when using this visual signal, namely saccadic inhibition and the RDE. At this stage, three hypotheses could explain this difference: (i) the modulation resulting from cue combination is specific to audiovisual integration; (ii) sensory cues have to come from different modalities to combine; (iii) the properties of the irrelevant visual stimulus used in experiment C do not permit the observation of visual-visual integration. These questions are addressed in the experiments reported below. More specifically, in experiment D we replaced the transient change in background luminance used in experiment C with a less visible pair of isoluminant strips presented in peripheral vision to avoid overlapping with the attended locations where targets are displayed. Indeed, the high saliency of the visual signal used in experiment C could be responsible for strong masking effects, thereby preventing the observation of subtle visual-visual interactions. Experiments E and F investigated whether a tactile signal presented around the onset of the visual target might produce a modulation of saccade latency similar to the one observed with auditory stimuli.

## Experiment D–Strips flashes & saccades

The change in background luminance used in experiment C was particularly salient. Notably, the whole background luminance was significantly altered for 10 ms over a large portion of the visual field, which included the attended regions of the visual field where the visual target was presented. This salient signal could have generated a strong saccadic inhibition resulting in the significant masking effects observed in experiment C, behind which visual-visual integration evidence would be concealed. To rule out this explanation, we created a new visual signal where the overall changes in saliency were much reduced. The background flash was replaced by a pair of horizontal isoluminant strips flashed for 10 ms in peripheral vision (see **Fig 1**). The strips displayed in the top and bottom of the screen (10.5˚ to 15.1˚ vertical eccentricity) had an average luminance equal to the grey background (50% corresponding to 25.8 cd/m$^2$). The experimental protocol was otherwise similar to the one in experiment C. 4 volunteers (2 women and 2 men) participated in this experiment, all from experiment C's pool of subjects. They were aged between 36 and 43 years old (average 39.8), all had normal or corrected-to-normal visual acuity and all were right handed.

### Results

A total of 3780 out of 3850 recorded trials were analyzed (98.4%). Only one saccade going in the wrong direction was not removed by the adaptive low-pass filter, which shows that the vast majority of anticipatory saccades were correctly eliminated. Given that saccade latency of leftward and rightward saccades did not differ–no main effect ($F(1,3) = 0.022$, $p = 0.89$) nor interaction with SOA ($F(8,24) = 1.19$, $p = 0.34$)–we pooled them together for further analyses. Latency n$_{scores}$ were computed for each of the 9 SOA conditions taking the *No flash* condition as a baseline. **Fig 5** summarizes the results of experiment D, showing the effect of strips flashes on saccadic onset (left panel) and comparing it with the effect of background flashes reported in experiment C (right panel). SOA had a significant main effect on the n$_{scores}$ ($F(8,24) = 4.56$, $p<0.002$, $\eta_p^2 = 0.6$). Student t-tests comparing the SOA = 0 ms reference condition with each of the other SOA condition (see **S1 Table**) revealed that flashes occurring just before the target tended to increase latencies (p = 0.038 and p = 0.031 for SOA = −60 and −30 ms, respectively), a tendency that became mostly significant with flashes presented after the target (p<0.0026 for

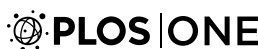

**Fig 5. Experiment D. Effect of strips flashes on saccade onset (left panel).** Median latencies averaged across participants for each SOA condition (top) and the corresponding $n_{scores}$ (bottom). Dashed lines show the *No flash* condition level and grey lines show individual results. Error bars indicate inter-individual SEM. Statistics included a single sample t-test performed on the $n_{scores}$ of the SOA = 0 ms reference condition to highlight the difference with the *No flash* condition (black arrow), and paired t-tests comparing this reference with each of the other SOA condition (red stars for each SOA above the X-axis). **Background flashes vs. strips flashes (right panel).** Comparison between experiment D (red plot) and C (grey plot): median latency differences (top) and $n_{score}$ differences (bottom) aligned on the *No flash* condition (dashed lines). Statistics were unpaired t-tests performed on the $n_{score}$ differences between experiments for each of the 9 SOA conditions. Three stars indicate highly significant differences after Bonferroni correction (p<0.00555) while single stars significant differences without correction (p<0.05). Red arrows indicate the overall range of variations.

SOA = +30 and +120 ms, and p = 0.018 for SOA = +60 ms). This impairment is partly confirmed by the reduced accuracy observed in the landing positions when flashes were delivered after target onset: saccadic gains tended to be lower (p = 0.012 for SOA = +120 ms, see supporting information **S2 Fig** right). Curiously, a single sample t-test on the $n_{scores}$ of the SOA = 0 ms condition showed that adding synchronous flashes tended to reduce saccade latencies by 9.9 ms (p = 0.020), although across the whole range of SOAs, flashes did not affect latencies (p = 0.51).

In order to compare the results of experiment D with the results of experiment C, we computed for both experiments the individual median latency differences between each SOA condition and its respective *No flash* baseline. We then performed unpaired Student t-tests to compare these latency differences between experiments (see **S3 Table**) and found a quite similar pattern of behavior, except that with strips flashes saccades were significantly faster when SOA = 0 ms (p<0.0055), and the limited warning effect found for SOA = −120 and −60 ms with background flashes tended to disappear (p = 0.041 and p = 0.057, respectively).

## Discussion

In this experiment, we used a new visual signal–isoluminant strips flashed for 10 ms in peripheral vision–designed to induce a much lower transient change in saliency than the full-background luminance decrease used in experiment C. During the debriefing, subjects reported that these flashes were indeed barely visible. We expected that this type of visual signal would limit the remote distractor effect (RDE), and thus reveal a potential modulation–even limited–of saccade latency similar to the one observed with the auditory signal. The effects on saccade latency were to some extent reduced with strips flashes compared to background flashes, which supports our hypothesis regarding the limited RDE. The limited warning effect that we observed when the background flashes were presented 120 ms and 60 ms before the target onset was eliminated with the strips flashes. However, there was still no evidence for visual-visual temporal interactions: irrelevant flashes did not combine with the target onset to modulate saccade latencies. The strips flashes also led to a reduced accuracy in saccade landing positions when the flash occurred after target onset. Interestingly, when the target and signal onset were synchronous, saccade latencies were shorter. As previously discussed, the target visibility could be increased by the simultaneous flash, producing an effect opposite to the RDE. One might speculate that, although with the low-saliency signal, this effect could have been reduced, this was not the case and since the masking effect mediated by the saccadic inhibition was reduced, the combined effect resulted in saccade latencies 10 ms shorter than without visual signal. To summarize, contrary to what we observed with auditory signals, visual targets were not temporally integrated to visual signals, and consequently did not modulate saccade latencies. This multisensory effect is therefore either exclusive to audiovisual interactions, or requires a sensory signal coming from another, non-visual, modality. To address this question, experiments E and F investigated whether we could evidence multisensory interaction with tactile and audio-tactile signals.

## Experiment E–Touches & saccades

We found that visual signals–whether salient or not–presented around target onset do not produce the same temporal interaction resulting in saccade latencies modulation as auditory signals do. In a new experiment, we replaced the auditory event (beep) with a brief tactile stimulation (100Hz vibration for 10 ms) to investigate whether a similar multisensory integration effect could be observed. Participants were asked to place their hands on the table at a comfortable distance. One hand was placed on top of the other, and the tactile stimulation was delivered at the center of the dorsal surface of the top hand by a solenoid tactor (Dancer Design, further technical details available on manufacturer's webpage: http://www.dancerdesign.co.uk/products/tactor.html). The tactor was held in place by an adhesive ring and was controlled with Matlab. In order to avoid any possible biases induced by the hand receiving the tactile stimulation, the left and right hand positions were alternated after each block of 60 trials, hence stimulating alternately each hand. Closed headphones were used to prevent the participant from hearing the noise produced by the tactile vibrations. Except for the tactile stimulation and hand alternation, the experimental protocol was the same as for experiment A. 8 volunteers (5 women and 3 men) participated in this experiment, only one from the same initial pool. They were between 22 and 42 years old (average 29.6), all had normal or corrected-to-normal visual acuity and all but one were right handed.

### Results

A total of 7520 out of 7680 recorded trials were analyzed (97.9%). Only 24 saccades going in the wrong direction were not removed by the adaptive low-pass filter, showing that the

majority of anticipatory saccades were eliminated. Latency $n_{scores}$ were computed for each of the 36 combinations of hand touched × target direction × SOA, taking the *No touch* condition of each hand touched × target direction as baselines. Saccade latencies did not differ when the tactile stimulation was delivered to the left or to the right hand–no main effect ($F(1,7) = 0.26$, $p = 0.63$); and the hand stimulated did not introduce a lateral bias for leftward or rightward saccades–no target direction × hand touched interaction ($F(1,7) = 0.85$, $p = 0.39$). Consequently, latencies observed in the left and the right hand conditions were pooled together for further analyses. Latency $n_{scores}$ were computed for each of the 18 combinations of target direction × SOA, taking the *No touch* condition of each direction baselines. The latency of leftward and rightward saccades did not differ–no main effect ($F(1,7) = 1.53$, $p = 0.26$) nor interaction with SOA ($F(8,56) = 0.32$, $p = 0.95$). Consequently, leftward and rightward saccades were pooled together for further analyses. Latency $n_{scores}$ were finally computed for each of the 9 SOA conditions, taking the *No touch* condition as a baseline. **Fig 6** summarizes the results of experiment E, showing the effect of tactile stimulation on saccade latency (left panel) and comparing it with the effect of beeps reported in experiment A (right panel). SOA had a significant main effect on the $n_{scores}$ ($F(8,56) = 41.08$, $p<0.0001$, $\eta_p^2 = 0.85$). Single sample t-tests on the $n_{scores}$ showed that saccade latencies were not reduced in the SOA = 0 ms condition with synchronous tactile stimulations ($p = 0.11$), although across the SOA range tactile stimulation tended to reduce latencies by 5.8 ms on average ($p = 0.044$), compared to the *No touch* baseline. Further paired t-tests comparing the SOA = 0 ms with each other SOA condition, showed that latencies decreased when touch was delivered before target onset ($p<0.001$ for SOA = −60 ms and below; and $p = 0.048$ for SOA = −30 ms) and increased when delivered after target onset ($p = 0.0082$ and $p<0.0002$ for SOA = +30 and +60 ms respectively).

In order to compare with the results of experiment A, we computed the individual median latency differences between each of the SOA condition and the *No beep* (experiment A) or *No touch* (experiment E) baselines. We then performed unpaired Student t-tests to compare these latency differences between experiments (see **S3 Table**) and found a very similar pattern of behavior, albeit shifted in time by roughly 30 ms. The tactile modulation found in the synchronous condition would correspond to the auditory modulation in the SOA = +30 ms condition. As a consequence, the warning effect with the tactile stimulation is to be found in the SOA = -30 ms, which explains why the SOA = 0 ms condition was not different than the *No touch* baseline.

## Discussion

The current experiment shows that the multisensory integration observed with experiment A and B are not specific to audiovisual interactions. Indeed, irrelevant tactile stimuli can significantly reduce saccade latencies in a similar way as irrelevant auditory stimuli. The fact that we did not observed a similar modulation with irrelevant visual signals, whether salient (background flash), or not (strips flashes) supports the idea that these interactions more likely happen between but not within modalities. Indeed, adding a brief irrelevant tactile stimulation around the time of target onset modulated the saccade reaction time so that earlier touches produced reduced latencies compared to later touches or no touch. Although the latency modulation pattern produced by the additional tactile stimulation appears very similar to the ones found with auditory beeps, the overall extent of this modulation was significantly reduced. Moreover, while auditory beeps delivered simultaneously or before target onset resulted in shorter latency saccades, only tactile stimulation delivered more than 30 ms before target onset resulted in shorter latencies. This could result from the different transduction delays associated to auditory and tactile signals, which could produce a shift in time for their respective effects

**Fig 6. Experiment E. Effect of tactile stimulation on saccade onset (left panel).** Median latencies averaged across participants for each SOA condition (top) and the corresponding $n_{scores}$ (bottom). Dashed lines show the *No touch* condition level and grey lines show individual results. Error bars indicate inter-individual SEM. Statistics included a single sample t-test performed on the $n_{scores}$ of the SOA = 0 ms reference condition to highlight the difference with the *No touch* condition (black arrow), and paired t-tests comparing this reference with each of the other SOA condition (red stars for each SOA above the X-axis). **Auditory beeps vs. tactile vibration (right panel).** Comparison between experiment E (red plot) and A (grey plot): median latency differences (top) and $n_{score}$ differences (bottom) aligned on the *No signal* condition (dashed lines). Statistics were unrelated t-tests performed on the $n_{score}$ differences between experiments for each of the 9 SOA conditions. Three stars indicate highly significant differences after Bonferroni correction (p<0.00555) while single stars and two stars significant differences without correction (p<0.05 and p<0.01 respectively). Red arrows indicate the overall range of variations.

on saccade latencies. Indeed, tactile stimulation on the hand has to reach the brain to be integrated with the visual stimulation, and this might be estimated to take approximately 25 ms–synaptic delays added to the impulse travel time in the ulnar sensory nerve with a conduction velocity of 48–74 m/s [47]. In comparison, auditory stimulation reaches the brain in just a few milliseconds. In other words, sounds presented through headphones would be transduced sooner than tactile stimuli delivered to the hand, which can account for the fact that saccades had shorter latencies when beeps but not tactile stimuli were simultaneous with the onset of the visual target. Experiment F will explore this issue in more details.

## Experiment F–Touches, beeps & saccades

In this last experiment, we directly compared the modulation of saccade latencies produced by auditory, tactile and audio-tactile signals. Contrasting the data from experiments A (auditory stimuli) and E (tactile vibrations) suggests that touch led to a smaller modulation of saccade latencies than auditory stimuli. Experiment F aimed at further investigating this issue using an

interleaved-modality within-subject design. This experiment also evaluated whether multi-modal irrelevant stimuli presented around the time of the visual target onset modulate saccade latencies more strongly compared to unimodal irrelevant signals. As reported above, the multi-sensory modulation of saccade latency resulting from tactile vibrations and auditory beeps are shifted in time relative to one another. This temporal shift explains why delivering beeps simultaneously with the visual target significantly decreased saccade latency but not touches. This difference might depend on the fact that sound are transduced faster than tactile stimuli delivered to the hand. Consequently, we conducted a preliminary experiment with a single participant to assess the temporal offset between the auditory and tactile stimuli that is required to perceive the two events simultaneously, the audio-tactile Point of Subjective Simultaneity (PSS). The PSS was estimated using a temporal order judgment task ("Touch was first/second?"). This experiment showed that a brief touch had to be delivered 28ms before the beep to be perceived as occurring simultaneously with the beep. This task ended up being extremely demanding and, although we tried to, we were unable to measure the correct values of this delay for each participant.

Based on the results obtained in this preliminary experiment, in experiment F, we used for all participants a fixed 30 ms delay between tactile and auditory stimuli (i.e. touch delivered 30 ms before beeps). This delay was also used in a tactile only condition: the tactile vibration that in experiment E occurred simultaneously with the onset of the visual target now occurred 30 ms before. 32 conditions were defined according to the following experimental design: [3 modality (tactile, auditory, audio-tactile) × 5 SOA (−120, −60, 0, +60, 120 ms) + No signal] × 2 touch hand (left, right). The touched hand was alternated in blocks of 64 trials, each with 4 repetitions per condition. 6 volunteers (4 women and 2 men) participated in this experiment. They were between 22 and 42 years old (average 29.5), all had normal or corrected-to-normal visual acuity and all but one were right handed.

## Results

A total of 7526 out of 7680 recorded trials were analyzed (98.0%). Once early saccades were excluded by our adaptive low-pass filter, only 15 saccades going in the wrong direction remained, which shows that the vast majority of anticipatory saccades have been removed. Leftward and rightward saccades being not different–no main effect ($F(1,5) = 0.012$, $p = 0.91$) nor interaction with SOA ($F(4,20) = 1.02$, $p = 0.42$) or modality × SOA ($F(8,40) = 0.53$, $p = 0.83$)–they were pooled together for further analyses. Latency $n_{scores}$ were computed for each of the 30 combinations of modality × SOA × hand touched, taking the corresponding *No signal* condition as baseline. Touching the left or right hand was not significantly different either–no main effect ($F(1,5) = 5.65$, $p = 0.063$, $\eta_p^2 = 0.53$) nor interaction with SOA ($F(4,20) = 0.56$, $p = 0.69$) or modality × SOA ($F(8,40) = 0.64$, $p = 0.74$)–they were also pooled together. Latency $n_{scores}$ were finally computed for each of the 15 combinations of modality × SOA, taking the respective *No signal* condition as baseline. The left plots of **Fig 7** show the effect of tactile, auditory and audio-tactile stimulation on saccadic median latency and its corresponding $n_{scores}$. SOA had a significant main effect on the $n_{scores}$ ($F(4,20) = 113.56$, $p<0.0001$, $\eta_p^2 = 0.96$), but not the modality ($F(2,10) = 1.16$, $p = 0.35$) although their interaction was significant ($F(8,40) = 6.17$, $p<0.0001$, $\eta_p^2 = 0.55$). For each modality, single sample t-tests on the $n_{scores}$ of the SOA = 0 ms reference condition (see **S2 Table**) showed a significant reduction of latencies of 8.3ms (p<0.006), 11.6ms (p<0.0005) and 14.1ms (p<0.0007), when adding a synchronous touch, beep or both, respectively. Further paired t-tests revealed that synchronous audio-tactile stimulations significantly decreased saccade latencies compared to when participants were presented with synchronous tactile stimuli (p<0.015; top-right plot of **Fig 7**). The other

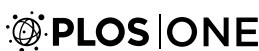

**Fig 7. Experiment F. Effect of tactile, auditory and audio-tactile stimulation on saccade onset. (Left panel)** Median latencies averaged across participants for each SOA condition (top) and their corresponding n$_{scores}$ (bottom). Dashed lines show the *No signal* condition level. Statistics performed on the n$_{scores}$ for each modality separately included a single sample t-test to compare the SOA = 0 ms reference condition with the *No signal* condition (colored arrows), and paired t-tests comparing this reference with each SOA condition (colored stars for each SOA above the X-axis). **(Top-right)** Average signal effect on latency n$_{scores}$ for each modality when SOA = 0 ms. **(Bottom-right)** Average slopes of the n$_{scores}$ linear regressions computed for each modality within the range [−60 ms, +60 ms]. Grey lines show individual results and error bars indicate inter-individual SEM. Comparisons were done using paired t-tests. Three stars indicate highly significant differences after Bonferroni correction (p<0.01) while single stars significant differences without correction (p<0.05).

comparisons between sensory modalities did not reach significance (p = 0.11 for tactile vs. audio and p = 0.27 for audio vs. audio-tactile).

Experiment F replicated the findings we observed in experiment A, B and E. Notably, auditory and tactile signals decreased and increased saccade latencies depending on whether they were presented before or after the visual target, respectively. Moreover, the fact that simultaneous beeps and tactile stimuli decreased saccade latencies suggests that this modulation was combined with a warning effect caused by the mere presentation of another sensory event. It is noteworthy that the injected delay introduced between auditory and tactile signals cancelled the temporal shift between modalities, as illustrated by the horizontal alignment of the latency modulation curves. However, these curves show different vertical scaling. In order to compare

the strength of the modulation between unimodal and multimodal signals, we computed individual linear regressions of $n_{scores}$ for each modality condition, within the range [−60 ms, +60 ms] where latency modulations were maximal (bottom-right plot of Fig 7). Average slopes (mean ± SEM in s$^{-1}$) were respectively 11.54 ± 1.32, 18.78 ± 1.84 and 21.63 ± 1.10 for tactile, auditory and audio-tactile stimulations. Paired Student t-tests revealed that slopes for tactile were lower than for auditory (p<0.04) and audio-tactile (p<0.01), but they were not significantly lower for auditory than for audio-tactile (p = 0.22).

Finally, the saccade latency spread quantified by the median absolute deviation followed a very similar pattern for all sensory signals, with a tendency to increase with SOA (SOA = +120 ms against SOA = 0 ms: p = 0.013 for auditory and p = 0.04 for audio-tactile). These results are consistent with what was observed in experiment A for auditory stimuli and in experiment E for tactile stimuli. We also tested whether for the audio-tactile signal, the auditory and tactile components were combined following the Bayesian optimal law predicted by the maximum likelihood estimator [MLE, see 48]. We computed for each participant the MLE prediction on the standard deviation of the saccade latencies for the audio-tactile combination using the following formula:

$$\hat{\sigma}_{AT}(0) = \frac{\sigma_A(0) \cdot \sigma_T(0)}{\sqrt{\sigma_A(0)^2 + \sigma_T(0)^2}} \tag{2}$$

where $\sigma_A(0)$ and $\sigma_T(0)$ are respectively the auditory and tactile standard deviations of the latency distributions for the SOA = 0 ms condition. We then compared this prediction $\hat{\sigma}_{AT}(0)$ with the standard deviation measured in the audio-tactile condition $\sigma_{AT}(0)$ using a paired student t-test (see supporting information S3 Fig). The MLE prediction was significantly lower than the observed one (p<0.001), indicating that the variability reduction expected in a Bayesian framework by combining two modalities is far from achieved. In fact, the audio-tactile bimodal variability was not even significantly lower than the ones for unimodal conditions (p = 0.93 for auditory and p = 0.49 for tactile).

## Discussion

Whether we add an irrelevant beep, touch or a combination of both, saccade reaction times are altered in similar ways. Compared to the *No signal* condition, latencies are generally shorter, and increase rather linearly for irrelevant stimuli presented in the range of [−120 ms; +120 ms] after the target onset. The within participant design of this experiment allowed us to confirm that auditory signals do produce a significantly stronger modulation effect than tactile signals, as we suggested when comparing experiments A and E. However, combining both modalities did not systematically increase the slopes of saccade latency modulations as a function of the SOA: audio-tactile signals produced a stronger modulation of saccade latencies than unimodal tactile signals, but only marginally when compared to the effects of unimodal auditory signals. This suggests that the saccade latencies reached a lower limit with auditory modulation with a saturation that did not allow further processing improvement with an additional modality, e.g. tactile. Finally, in many spatial perceptual problems the multisensory cue-combination is found to be optimal according to MLE Bayesian framework, for example when using both haptic and visual modalities to estimate size [48]. In the temporal domain, such optimality has been debated, some studies advocating for optimal audiovisual combination [49], while others refuted it [4,50,51]. For saccade triggering, the combined interference of irrelevant auditory and tactile signals with latency clearly did not follow Bayes law, which points to potentially distinct underlying integration mechanisms than for perception.

Alternatively, for visually guided saccades the brain might bypass the slowest tactile signal and keep only the more precise auditory signal to interact with the target onset [8]. Because saccades involve specific brain structures–the superior colliculus–one cannot exclude that the integration of the different sensory maps could follow different rules than in other goal-directed motor actions.

## General discussion

### A multisensory effect between modalities but not within

In this project, we conducted behavioral studies to address a key issue in the multisensory field: is there a supramodal system in the brain which gathers perceptual evidence from the available sensory inputs in order to guide our behavior? Our experiments showed that irrelevant auditory signals (beeps) presented around target onset influenced oculomotor reactions, with a temporal modulation that depends on the SOA between the target and the irrelevant signal. Notably, saccade latencies decreased or increased when the visual target onset was preceded or followed by a task-irrelevant auditory stimulus (experiments A and B). A similar modulation was observed also when auditory stimuli were replaced by or combined with tactile vibrations delivered to participants' hand (experiments E and F). These results suggest that the visual target onset was temporally bound to the task-irrelevant auditory and/or tactile signals. Consequently, participants might have perceived the onset of the visual target sooner or later, depending on whether auditory and tactile signals preceded or followed the target, which, ultimately, modulated saccade latencies.

However, we did not observe the same latency modulation when using transient visual signals (experiments C and D). In fact, we found that the visual signals often increased saccade latencies even when they occurred before target onset. This effect can be attributed to saccadic inhibition and the related remote distractor effect rather than sensory integration processes. In line with our predictions, saccadic inhibition appears to be stronger for the salient signal (black background, experiment C) than for the weaker signal (isoluminant strips, experiment D). Interestingly, when the flash and the target onset occurred simultaneously, the target visibility increased which compensated for the saccadic inhibition latency impairment. In sum, our findings show that the modulation of oculomotor response is not specific to audiovisual interactions, as touch can also influence saccade reaction times. However, we did not find such modulation with visual signals, which indicates that external signals only contribute between and not within modalities.

### A potential illusory overlap/gap effect

It is puzzling that the extent latency modulation reported here when using audiovisual stimulations (respectively 11 and 13.5 ms in experiment A and B) was far weaker than what has been reported when simply estimating the perceived time shift induced by multisensory integration of audiovisual events [about 120 ms, 4]. We propose that this difference might be due to a perceptual illusion that limited the effects on the saccadic latencies. During debriefing, some of the participants reported a peculiar temporal sequencing of the target onset and the fixation offset: although both events were always perfectly synchronous, when the beep occurred after the target onset, it seemed as if, for a brief instant, there was neither fixation nor target on the screen, as if a temporal gap had been introduced between the two events. This phenomenon points toward the possibility that only the perceived target onset was affected by some temporal multisensory integration mechanism [4–6]. The multisensory temporal integration could have shifted the target onset so that it is perceived later than the physical event for beeps after (positive SOA), producing an illusory gap. Conversely, the relative target onset would be

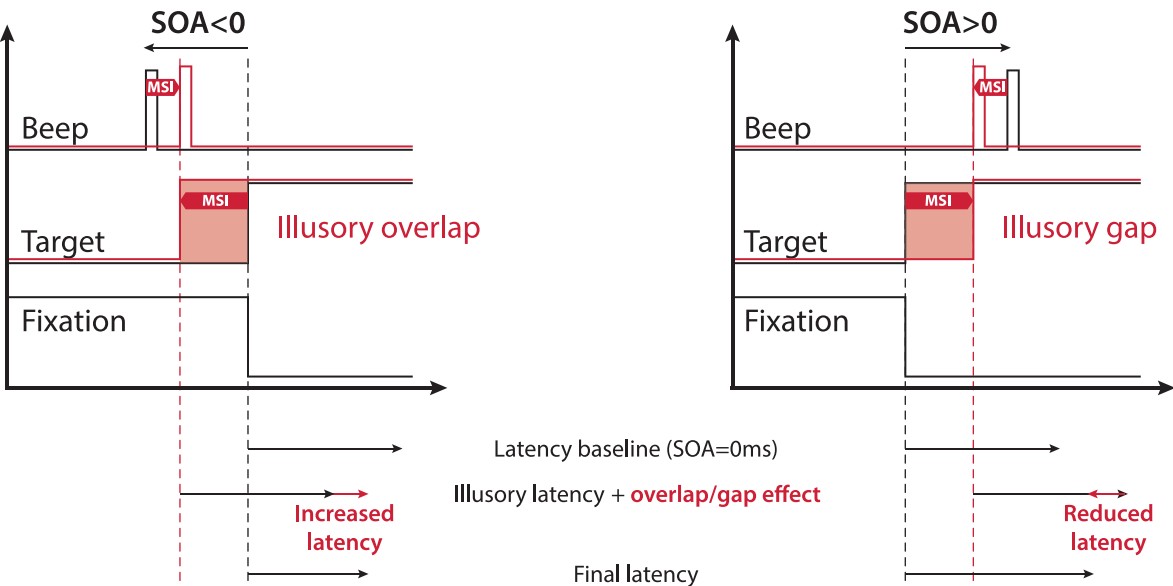

**Fig 8. Saccade latency components.** Illustration of how temporal multisensory integration could introduce an illusory gap/overlap for positive/negative SOA. This additional effect would have reduced the extent of the multisensory modulation of saccade latencies by reducing the effect of the perceived target onset time shift.

perceived sooner for beeps first (negative SOA), producing an illusory overlap (illustrated in **Fig 8**).

In both situations, the expected modulation by the audiovisual integration would be reduced by the illusory gap/overlap, as the latter goes in opposite direction than the multisensory effect. Indeed, early studies investigated the effects of an overlap/gap paradigm where the fixation offset and target onset where manipulated separately so that a gap (neither on the screen) or an overlap (both on the screen) was introduced [21,52]. Saccade latencies are greatly reduced with a gap and increased with an overlap. Importantly these effects are also present for saccades toward acoustic targets, which suggests that the facilitation of premotor processes in the superior colliculus could be responsible for the gap effect rather than the processing of the target itself [53,54]. We speculate that an illusory gap/overlap due to beeps shifting the perceived onset time of the target–could have minimized the effect of beeps on saccades. This interesting issue is currently under investigation.

## Conclusion and open questions

We found that both auditory and tactile events falling within a large binding window of visual events reliably affect saccadic latencies. This modulation is stronger when auditory and tactile signals are combined. On the contrary, an external visual signal mostly impairs saccadic reactivity. This demonstrate that combining multiple sources of information does not systematically lead to improved performance, as often reported in the multisensory literature. Mechanisms specific to each motor behavior might have different ways of processing and combining those sources of information. For saccades, auditory and tactile information do interact to improve the reactivity to a visual target onset, but not visual information. To illustrate the functional significance of this phenomenon, let us consider an ecological example: while walking on grassy slopes, careless bees flying around us or grasshoppers jumping as we approach can produce both auditory and tactile signals. These signals combine with the visual

signal to improve the localization about where to aim the saccade and quickly foveate the moving insect. Here we showed that even when the sound and/or touch are non-informative about where the target is located and when the event occurs, saccades are still triggered faster. One might wonder whether this ability could have an evolutionary grounding. Reacting to something moving in the visual field and making noise would be more crucial for survival than if moving silently, or is it just a matter of general alertness?

## Supporting information

**S1 Table. Student's t-tests and related tests for all experiments but F.** The statistical analyses to compare the mean median latency $n_{score}$ of each condition with the SOA = 0 ms reference condition followed 3 steps: a normality test (Shapiro-Wilk), a variance test (Levene) and a paired Student's t-test. This table reports the outcome of all these tests together with the effect size (Cohen's d).
(XLSX)

**S2 Table. Student's t-tests and related tests of experiment F.** The statistical analyses to compare the mean median latency $n_{score}$ of each condition with the SOA = 0 ms reference condition followed 3 steps: a normality test (Shapiro-Wilk), a variance test (Levene) and a paired Student's t-test. This table reports the outcome of all these tests together with the effect size (Cohen's d).
(XLSX)

**S3 Table. Student's t-tests and related tests for comparisons between experiments.**
(XLSX)

**S1 Fig. Saccades going in the wrong direction of experiment A.** Individual histograms plotting the number of initial saccades going in the opposite direction compared to where the target appeared for each SOA and *No beep* conditions. The **left panel** shows the initial data with all the correctly detected saccades and the **right panel** shows the data after removing those with a gain below 0.4 or with latencies below 80 ms or above 400 ms.
(TIF)

**S2 Fig. Effect of background flashes (left) and strips flashes (right) on saccadic gain.** Saccadic gain averaged across participants for each SOA condition (top) and the corresponding $n_{scores}$ (bottom). Dashed lines show the *No flash* condition level and grey lines show individual results. Error bars indicate inter-individual SEM. Statistics included a single sample t-test performed on the $n_{scores}$ of the SOA = 0 ms reference condition to highlight the difference with the *No flash* condition (black arrow), and paired t-tests comparing this reference with each other SOA condition (red stars for each SOA above the X-axis).
(TIF)

**S3 Fig. Combination of auditory and tactile signals not Bayesian optimal.** Average latency standard deviation computed when SOA = 0 ms for each modality with the associated MLE optimal prediction. Comparisons were done using paired t-tests.
(TIF)

## Acknowledgments

The authors are grateful to Françoise Vitu for fruitful discussions and suggested relevant literature. This research was in part funded by the Agence Nationale de la Recherche (grant number

ANR-18-CE10-0001) awarded to Andrea Desantis. The funders had no role in study design, data collection and analysis, decision to publish, or preparation of the manuscript.

## Author Contributions

**Conceptualization:** Manuel Vidal, Laurent Madelain.

**Funding acquisition:** Andrea Desantis.

**Investigation:** Manuel Vidal.

**Methodology:** Manuel Vidal, Andrea Desantis, Laurent Madelain.

**Project administration:** Manuel Vidal.

**Resources:** Manuel Vidal.

**Software:** Manuel Vidal.

**Validation:** Manuel Vidal.

**Visualization:** Manuel Vidal, Laurent Madelain.

**Writing – original draft:** Manuel Vidal.

**Writing – review & editing:** Manuel Vidal, Andrea Desantis, Laurent Madelain.

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
