## [Decision Letter · Decision Letter 0]

13 Dec 2019

PONE-D-19-21342

Irrelevant auditory and tactile signals interact with the visual target onset and modulate saccade latencies, not visual signals

PLOS ONE

Dear Dr. Vidal,

Thank you for submitting your manuscript to PLOS ONE. After careful consideration, we feel that it has merit but does not fully meet PLOS ONE’s publication criteria as it currently stands. Therefore, we invite you to submit a revised version of the manuscript that addresses the points raised during the review process.

We would appreciate receiving your revised manuscript by Jan 27 2020 11:59PM. To enhance the reproducibility of your results, we recommend that if applicable you deposit your laboratory protocols in protocols.io, where a protocol can be assigned its own identifier (DOI) such that it can be cited independently in the future. For instructions see: http://journals.plos.org/plosone/s/submission-guidelines#loc-laboratory-protocols

We look forward to receiving your revised manuscript.

Kind regards,

Nicholas V Swindale

Academic Editor

PLOS ONE

Journal Requirements:

2. Thank you for including your ethics statement: Participants gave a written consent after being informed of the experimental methods used and their right to interrupt the experiment at any time. The whole project was approved by the local ethics committee and complies with the regulations described in the Declaration of Helsinki.

Additional Editor Comments (if provided):

Dear Dr. Vidal,

I have now received two reviewers' reports on your manuscript "Irrelevant auditory and tactile signals interact with the visual target onset and modulate saccade latencies, not visual signals". Both reviewers agree that overall your conclusions are technically sound and the manuscript is well written. I am happy to invite you to revise and resubmit the manuscript in accordance with the suggestions of both reviewers. Could you also address the issue of whether data will be shared in accordance with the PLOS data policy?

Apologies once again for how long it has taken to reach this point.

Sincerely,

Nicholas Swindale

Reviewers' comments:

Reviewer's Responses to Questions

**Comments to the Author**

1. Is the manuscript technically sound, and do the data support the conclusions?

Reviewer #1: Yes

Reviewer #2: Yes

2. Has the statistical analysis been performed appropriately and rigorously? 

Reviewer #1: Yes

Reviewer #2: Yes

3. Have the authors made all data underlying the findings in their manuscript fully available?

Reviewer #1: Yes

Reviewer #2: No

4. Is the manuscript presented in an intelligible fashion and written in standard English?

Reviewer #1: Yes

Reviewer #2: Yes

5. Review Comments to the Author

Reviewer #1: There is an overall scarcity of research papers addressing the processing of multisensory signals. In this contribution, the authors use saccadic eye movements as a model system to study the facilitatory or modulatory role of auditory and tactile signals on visual signal processing, reporting the following key findings: (1) Simultaneous presentation of auditory / tactile signal and visual target resulted in a “warning effect” and shorter saccade latencies. (2) When auditory / tactile signals were not synchronized with visual target presentation, a “modulation effect” occurred which could result in either shorter or longer latencies; a visual signal resulted in increased latencies. (3) A combination of both modalities increased the modulation effect only to a small extent, interpreted as a saturation of the multisensory interaction with oculomotor control. This paper makes an important and novel contribution to our understanding of how we integrate information from different senses to quickly direct the eyes to objects of interest. The paper is well written and overall technically sound, but I have some concerns about the size of the obtained effects based on the small sample size.

Main results for exp. 1 rely on data that are relatively diverse across the small tested subject population (see Fig. 2 upper panels), yet a highly significant latency difference of 11 ms is reported. The authors also describe their findings as homogenous across observers, yet, Fig. 2 paints a different picture. It appears that for a simultaneous presentation there was no latency difference for at least three observers. These differences have to be acknowledged and discussed, and I’d like to see a more detailed analysis regarding different subgroups of observers, in case there are systematic differences. This comment also applies, perhaps to a lesser degree, to the other experiments.

p.2, para.1: the second half of this paragraph discusses temporal perception and even though this is interesting, it is somewhat unrelated to the rationale of the study. In fact, I found it misleading because it sets up the study as something that it does not deliver. There is no measure of perception here, the variable of interest is saccadic reaction time or latency. Whereas it could be argued that saccades are related to perception, there is also evidence that this is not always the case. I would much prefer to see this introduction lead more towards using eye movements as a model system of studying multisensory processing than as a measure of temporal perception (as nicely done on, p.3, para.2, and even more in para.3 on that page and on p.4). My suggestion would be to either rephrase or simply delete the paragraph in question on p.2.

p.7, Data processing: I am surprised that the standard Eyelink saccade detection algorithm was used. From my lab’s experience, this algorithm is not sensitive enough to detect small saccades. Given that saccades are the main dependent variable in this study, and that the authors are looking into sensitively detecting very early saccades as well, I’d like to see either a discussion and justification for using this algorithm, or a comparison of results obtained with a more sensitive algorithm. It is also somewhat unusual to apply a filter to saccade latencies and remove anticipatory / early onsets. I’d like to see a latency distribution. Whereas typical saccade latencies might be long (around 200 ms or more), there are also reports in the literature of short-latency or express saccades with latencies well below 100 ms, and the present paradigm might well have triggered those.

Overall results presentation: since the procedure and results across experiments are similar and reported in parallel fashion, it would be helpful to streamline the results section overall and report results for control or follow-up experiments in more abbreviated form.

Minor

p.3, l.13: correct in-text citations (Sparks without initials), see also l.22 and other places throughout the manuscript

p.5, l.6-7: rephrase sentence, sounds awkward (“because they provide… allowing the quantification of…”)

p.5, Participants: did participants have normal or corrected-to-normal visual acuity, and if yes, was this tested? Why is it important that subjects were right-handed? Did the results of the two authors differ in any way from those obtained from naïve participants? Were participants remunerated for participation?

p.7, l.1&3: replace “totalizing” by “totaling”

Figure 2 and all similar figures: axes labels are almost impossible to read. Please improve the general aesthetics and readability of this figure with larger font sizes and clearer labels.

p.18: stripes and not strips?

p.21, l.11: please provide additional detail on this device and its manufacturer

p.28, para.1, first sentence: I do not think that the presented results can address the question how this information is processed in the brain; results are purely behavioral and should not be overinterpreted

p.28, l.29: reported as 11 ms above

p.30, title: consider rephrasing “openings” to “limitations” or “open questions”

p.30, last sentence: this is a matter of taste, but this sentence does not add much in terms of content as it is speculative

Reviewer #2: Irrelevant auditory and tactile signals interact with the visual target onset and modulate saccade latencies, not visual signals

The authors present a series of systematic experiments to investigate the effect of visual, auditory and tactile stimuli on saccadic latencies. These additional and irrelevant stimuli are presented with a stimulus onset asynchrony relative to the onset of the visual saccadic target. The authors find that auditory and tactile stimuli can shorten saccadic reaction times if they are presented before, but slow down after. Visual stimuli mostly just increase the saccadic latency.

Generally this is a very well written paper, with a good number of well designed experiments. Comments are relatively minor.

Title: Irrelevant auditory and tactile signals, (but not visual signals), interact with the visual target onset and modulate saccade latencies, (not visual signals): not sure whether the visual signals need to be in title, but in its current form it is really confusing.

P2 Ln 14 and many others: There is a very relevant paper in integration of signals from different modalities by Crevecoeur, Munoz and Scott, J, Neurosci 2016 (Dynamic Multisensory Integration: Somatosensory Speed Trumps Visual Accuracy during Feedback Control). It is not the same, but I believe it adds to your story, also in the discussion.

P2 Ln 26: "famous illusions", this is a non-informative statement. Make the illusions explicit or remove.

P3 Ln 2-3: I understand the physics, but please explain more carefully how sound and vision of the balloon first move at very different speeds through the air and are transduced into the nervous system with different latencies. I had to read this sentence twice to understand. Maybe even a graph. Make distinction between physical travel of information and neuronal processing of that information.

Pg 6, Ln 7: The probability of an event happening is increasing over time, you do not have a flat hazard. How can this have affected your results? Subjects might have been anticipating a response.

Pg 6, Ln 14: Was the subject explicitly told that these events were irrelevant?

Pg 9, Ln 9 'than with a synchronous beep for SOA= -120 ms', I don't understand this statement. Please rephrase.

Fig 2, bottom panels, add latency and MAD to the n-score

Pg 10 ln 12: discuss causal inference here (e.g. Shams, 2010, Causal Inference in Perception). That is probably why effects go back to baseline.

Fig 3, bottom panels, add latency and MAD (Also in all other figures)

Pg 26, Ln3 and further: This multisensory integration and time can and should be discussed in light of the findings of Crevecoeur et al., (2016).

Pg 28, Ln 20: Curiously should be Interestingly

Pg 28, Ln 26: " ... has been reported in a different context" -> please say something about it. Now this sentence does not add any info.

Pg 29, Ln 5: remove a 'the'

Pg 29, Ln 15: these effects or this effect

6. PLOS authors have the option to publish the peer review history of their article (what does this mean?). If published, this will include your full peer review and any attached files.

Reviewer #1: No

Reviewer #2: No

---

## [Author Response · Author response to Decision Letter 0]

28 Dec 2019

Response to the Reviewers

General comments

3. Have the authors made all data underlying the findings in their manuscript fully available?

Reviewer #2: No

Data has been made readily available since the submission of the manuscript, but I forgot to state it also in the manuscript. I reorganized the folders in the repository so that it is easier to figure out what the folders and files contain, added the python processing code and it is now clearly stated it in the ‘Data processing’ section of the revised manuscript: ‘Raw data files, global plots, individual plots and data processing code for all experiments can be found in the following public repository:

https://amubox.univ-amu.fr/s/PFHTMzb6i7emqfc‘

Specific comments

Reviewer #1

There is an overall scarcity of research papers addressing the processing of multisensory signals. In this contribution, the authors use saccadic eye movements as a model system to study the facilitatory or modulatory role of auditory and tactile signals on visual signal processing, reporting the following key findings: (1) Simultaneous presentation of auditory / tactile signal and visual target resulted in a “warning effect” and shorter saccade latencies. (2) When auditory / tactile signals were not synchronized with visual target presentation, a “modulation effect” occurred which could result in either shorter or longer latencies; a visual signal resulted in increased latencies. (3) A combination of both modalities increased the modulation effect only to a small extent, interpreted as a saturation of the multisensory interaction with oculomotor control. 

This paper makes an important and novel contribution to our understanding of how we integrate information from different senses to quickly direct the eyes to objects of interest. The paper is well written and overall technically sound, but I have some concerns about the size of the obtained effects based on the small sample size.

Main results for exp. 1 rely on data that are relatively diverse across the small tested subject population (see Fig. 2 upper panels), yet a highly significant latency difference of 11 ms is reported. The authors also describe their findings as homogenous across observers, yet, Fig. 2 paints a different picture. It appears that for a simultaneous presentation there was no latency difference for at least three observers. These differences have to be acknowledged and discussed, and I’d like to see a more detailed analysis regarding different subgroups of observers, in case there are systematic differences. This comment also applies, perhaps to a lesser degree, to the other experiments.

In order to reduce dispersion between subjects, statistics were computed on the n-scores. This pre-processing stage removes the overall vertical offset variability between subjects which allows highlighting the strong regularity between subjects. It is important to point out that the ‘No beep’ baseline in the upper plots is the average value across subjects (it would be very confusing to add individual grey lines for each). Although this average might be inferior for 2 or 3 subjects, once taking their respective values the ‘No beep’ is always way above, as indicated in the n-score plot where the ‘No beep’ baseline is 0 for all subjects. This difference is indeed systematic and highly significant even after Bonferroni correction.

We would like to insist on the fact that we took great care in the way we conducted statistical analyses and we report the outcome of these analyses in great details in the manuscript. Individual data and standard errors are readily visible in the plots, ANOVAs were conducted prior to planned comparisons, effect sizes are provided, Student t-test for mean comparisons were computed after veryfing that the nscores did not deviate from normality and that they met the homoscedasticity assumption (Shapiro-Wilk and Levene test), and finally the alpha value for significance level has always been strictly corrected for multiple tests using Bonferroni (p<0.00555 most of the times).

p.2, para.1: the second half of this paragraph discusses temporal perception and even though this is interesting, it is somewhat unrelated to the rationale of the study. In fact, I found it misleading because it sets up the study as something that it does not deliver. There is no measure of perception here, the variable of interest is saccadic reaction time or latency. Whereas it could be argued that saccades are related to perception, there is also evidence that this is not always the case. I would much prefer to see this introduction lead more towards using eye movements as a model system of studying multisensory processing than as a measure of temporal perception (as nicely done on, p.3, para.2, and even more in para.3 on that page and on p.4). My suggestion would be to either rephrase or simply delete the paragraph in question on p.2.

This paragraph has been merged with the next and largely rewritten. According to the reviewer’s suggestion, we kept only minimal literature on multisensory temporal integration for perception which allows introducing how such perceptual biases can influence motor behavior (e.g. saccades).

p.7, Data processing: I am surprised that the standard Eyelink saccade detection algorithm was used. From my lab’s experience, this algorithm is not sensitive enough to detect small saccades. Given that saccades are the main dependent variable in this study, and that the authors are looking into sensitively detecting very early saccades as well, I’d like to see either a discussion and justification for using this algorithm, or a comparison of results obtained with a more sensitive algorithm. It is also somewhat unusual to apply a filter to saccade latencies and remove anticipatory / early onsets. I’d like to see a latency distribution. Whereas typical saccade latencies might be long (around 200 ms or more), there are also reports in the literature of short-latency or express saccades with latencies well below 100 ms, and the present paradigm might well have triggered those.

The reviewer is right: the standard Eyelink saccade detection algorithm is not so accurate to detect small saccades. However in our protocol the saccades we targeted had an amplitude of 8° which is not really small, and the (very few) smaller saccades were excluded from data processing according to an adaptive filter described in the ‘Data processing’ section of experiment A. In our study, saccades programmed before the visual target could have been processed could bias the results in the same direction as the findings we report. It is therefore crucial to remove all anticipatory saccades, which could be achieved by removing small amplitude saccades often related to a wrong start. We are aware of the existence of express saccades, and we provide a detailed argument about how we handle them in the second to last paragraph of experiment A’s discussion.

All the saccade distributions for every experiment, subject and experimental condition are readily available in the public data repository in the folder ‘Individual plots>Distribution analyses’.

Overall results presentation: since the procedure and results across experiments are similar and reported in parallel fashion, it would be helpful to streamline the results section overall and report results for control or follow-up experiments in more abbreviated form.

The reviewer is right, the method and data processing has been fully detailed in experiment A and in the following only differences with the first experiment have been described. We tried to keep the result sections of the following experiments as short as possible. We fear that reducing the result presentation would be done at a cost: hiding some of the statistics that we believe are important, as required in the PlosOne guidelines. Could the reviewer give us more precise suggestions of how we could limit the size of the result presentation?

Minor

p.3, l.13: correct in-text citations (Sparks without initials), see also l.22 and other places throughout the manuscript

Done

p.5, l.6-7: rephrase sentence, sounds awkward (“because they provide… allowing the quantification of…”)

Rephrased: ‘Saccadic eye-movements were chosen for they provide fast and reliable responses that allow measuring small differences in decision processing delays’

p.5, Participants: did participants have normal or corrected-to-normal visual acuity, and if yes, was this tested? Why is it important that subjects were right-handed? Did the results of the two authors differ in any way from those obtained from naïve participants? Were participants remunerated for participation?

Yes, participants had all normal or corrected-to-normal visual acuity: this has been added for each experiment. This was not tested, since most were students from the lab and we could rely on their judgment for visual acuity. Not all the participants were right-handed (see experiments B, E and F). Finally, since participating in experiments is part of the students’ duty in the lab, they were not remunerated for their time.

The authors’ results were always perfectly in line with the average results. The reviewer can look at the individual results provided in the public repository; the initials of the two authors who participated are MV and LM (https://amubox.univ-amu.fr/s/PFHTMzb6i7emqfc).

p.7, l.1&3: replace “totalizing” by “totaling”

Done

Figure 2 and all similar figures: axes labels are almost impossible to read. Please improve the general aesthetics and readability of this figure with larger font sizes and clearer labels.

This figure as well as all the other result’s figures will be displayed in a full page in the final publication and will therefore look quite larger than what they look in the manuscript. Moreover, vectorial plots will be provided so that the reader can – if needed – zoom into the figure to increase at will the size of the labels and plots. However, following the reviewer’s suggestion, in the revised manuscript I have increased the font size of all plots by about 30%. I tried to find a good tradeoff between font size and readability of the important amount contained in these plots, I hope this will suit the reviewer.

p.18: stripes and not strips?

Strips, corrected.

p.21, l.11: please provide additional detail on this device and its manufacturer

I added a footnote pointing to the manufacturer’s webpage. 

p.28, para.1, first sentence: I do not think that the presented results can address the question how this information is processed in the brain; results are purely behavioral and should not be overinterpreted

We have softened what could be seen as over interpretation of behavioral results. The opening sentence of the general discussion now reads ‘In this project, we conducted behavioral studies to probe a key issue in the multisensory field: is there a supramodal system in the brain gathering perceptual evidence from the available sensory inputs in order to guide motor reactions?’

p.28, l.29: reported as 11 ms above

Corrected, it now reads ‘respectively 11 and 13.5 ms in experiment A and B’.

p.30, title: consider rephrasing “openings” to “limitations” or “open questions”

Changed to ‘Open questions’

p.30, last sentence: this is a matter of taste, but this sentence does not add much in terms of content as it is speculative

A matter of taste indeed, I do believe that it fits well with the ‘open questions’ (previously ‘openings’) included in the title �.

Reviewer #2

The authors present a series of systematic experiments to investigate the effect of visual, auditory and tactile stimuli on saccadic latencies. These additional and irrelevant stimuli are presented with a stimulus onset asynchrony relative to the onset of the visual saccadic target. The authors find that auditory and tactile stimuli can shorten saccadic reaction times if they are presented before, but slow down after. Visual stimuli mostly just increase the saccadic latency.

Generally this is a very well written paper, with a good number of well-designed experiments. Comments are relatively minor.

Title: Irrelevant auditory and tactile signals, (but not visual signals), interact with the visual target onset and modulate saccade latencies, (not visual signals): not sure whether the visual signals need to be in title, but in its current form it is really confusing.

We followed the reviewer suggestion and moved the ‘visual signals’ closer to the sentence segment it relates to. The title now reads: ’Irrelevant auditory and tactile signals, but not visual signals, interact with the visual target onset and modulate saccade latencies’

P2 Ln 14 and many others: There is a very relevant paper in integration of signals from different modalities by Crevecoeur, Munoz and Scott, J, Neurosci 2016 (Dynamic Multisensory Integration: Somatosensory Speed Trumps Visual Accuracy during Feedback Control). It is not the same, but I believe it adds to your story, also in the discussion.

This is a very relevant reference that we were not aware of. We thank the reviewer for giving us the reference and added it to the revised manuscript at several points in the introduction, and in the discussions. 

P2 Ln 26: "famous illusions", this is a non-informative statement. Make the illusions explicit or remove.

The whole paragraph has been reshaped according to Reviewer #1’s suggestion, and the reference to these illusions removed.

P3 Ln 2-3: I understand the physics, but please explain more carefully how sound and vision of the balloon first move at very different speeds through the air and are transduced into the nervous system with different latencies. I had to read this sentence twice to understand. Maybe even a graph. Make distinction between physical travel of information and neuronal processing of that information.

In the submitted manuscript this was detailed in the previous paragraph (‘The resulting brief acoustic signal and salient visual change generated by the explosion are transmitted in the air and then transduced by our auditory and visual systems at different speeds.’). However the reviewer is right, it could be explained closer to this sentence. Based on the suggestion of Reviewer #1 to reduce the previous paragraph, these two paragraphs have been rewritten and merged into a single paragraph. Hopefully, this will have clarified the reviewer’s concern.

Pg 6, Ln 7: The probability of an event happening is increasing over time, you do not have a flat hazard. How can this have affected your results? Subjects might have been anticipating a response.

After target onset, and even more so after a sound beep (for negative SOA conditions), the longer the delay the higher is the probability of the target to appear. The reviewer is right, participants could anticipate the target event and program a saccade in either direction that would have reduced overall latencies more for negative SOA than for positive SOA. This issue is thoroughly discussed in the second paragraph of experiment A’s discussion. To prevent such bias on performance, we have suppressed anticipatory saccades using an adaptive filter, well described in the ‘Data processing’ section.

Pg 6, Ln 14: Was the subject explicitly told that these events were irrelevant?

Yes they were. We emphasized this point adding a new sentence in the procedure: ‘(subjects) were explicitly told that the beep was irrelevant to the task and not to pay attention to it’, which replaces the previous ‘regardless of the non-informative beeps’ now moved to Fig 1’s caption.

Pg 9, Ln 9 'than with a synchronous beep for SOA=-120 ms', I don't understand this statement. Please rephrase.

This sentence has been rephrased to clarify the baseline and the different comparisons.

Fig 2, bottom panels, add latency and MAD to the n-score (Also in all other figures)

Done.

Pg 10 ln 12: discuss causal inference here (e.g. Shams, 2010, Causal Inference in Perception). That is probably why effects go back to baseline.

This reference addresses a very interesting issue in the field of perception which is very relevant to the binding/not binding issue we deal with in this work. We cited this reference in the introduction and in the discussion as suggested by the reviewer.

Pg 26, Ln3 and further: This multisensory integration and time can and should be discussed in light of the findings of Crevecoeur et al., (2016).

Crevecoeur et al’s work has been added in the discussion of experiment F (not in the results). It provides an alternative explanation to why combining tactile and audition did not increase significantly the saccade latency modulation effect.

Pg 28, Ln 20: Curiously should be Interestingly

Done.

Pg 28, Ln 26: " ... has been reported in a different context" -> please say something about it. Now this sentence does not add any info.

We removed from the manuscript the reference to Illis, Ernst et al (2002) work as it is not really relevant to our work. It deals with mandatory fusion within vs. between senses. We thank the reviewer for asking more details about this ‘different context’, which made us realize that it is too far from ours.

Pg 29, Ln 5: remove a 'the'

Done.

Pg 29, Ln 15: these effects or this effect

Corrected, now reads ‘these effects’.

---

## [Decision Letter · Decision Letter 1]

9 Jan 2020

PONE-D-19-21342R1

Irrelevant auditory and tactile signals, but not visual signals, interact with the target onset and modulate saccade latencies

PLOS ONE

Dear Dr. Vidal,

Thank you for resubmitting your manuscript to PLOS ONE. We have received some further comments from one of the two reviewers and we invite you to submit a further revised version of the manuscript that addresses the points raised by this reviewer, in particular the recommendation to shorten the Results section.

We would appreciate receiving your revised manuscript by Feb 23 2020 11:59PM. To enhance the reproducibility of your results, we recommend that if applicable you deposit your laboratory protocols in protocols.io, where a protocol can be assigned its own identifier (DOI) such that it can be cited independently in the future. For instructions see: http://journals.plos.org/plosone/s/submission-guidelines#loc-laboratory-protocols

A rebuttal letter that responds to each point raised by the academic editor and reviewer(s). This letter should be uploaded as separate file and labeled 'Response to Reviewers'.A marked-up copy of your manuscript that highlights the further changes made to the revised (second) version. This file should be uploaded as separate file and labeled 'Revised Manuscript with Track Changes'.An unmarked version of your revised paper without tracked changes. This file should be uploaded as separate file and labeled 'Manuscript'.

We look forward to receiving your revised manuscript.

Kind regards,

Nicholas V Swindale

Academic Editor

PLOS ONE

Reviewers' comments:

Reviewer's Responses to Questions

**Comments to the Author**

1. If the authors have adequately addressed your comments raised in a previous round of review and you feel that this manuscript is now acceptable for publication, you may indicate that here to bypass the “Comments to the Author” section, enter your conflict of interest statement in the “Confidential to Editor” section, and submit your "Accept" recommendation.

Reviewer #1: (No Response)

2. Is the manuscript technically sound, and do the data support the conclusions?

Reviewer #1: Yes

3. Has the statistical analysis been performed appropriately and rigorously? 

Reviewer #1: Yes

4. Have the authors made all data underlying the findings in their manuscript fully available?

Reviewer #1: Yes

5. Is the manuscript presented in an intelligible fashion and written in standard English?

Reviewer #1: Yes

6. Review Comments to the Author

Reviewer #1: The authors have addressed my previous comments. My suggestion for the Results sections would merely be to try to be more concise. All Results sections are very wordy and could easily be shortened by at least 30% simply by reducing words (example: "differed" instead of "differed greatly").

Minor:

Exp. D: title still says "strips" instead of "stripes"; please proof-read this section

7. PLOS authors have the option to publish the peer review history of their article (what does this mean?). If published, this will include your full peer review and any attached files.

Reviewer #1: No

---

## [Author Response · Author response to Decision Letter 1]

10 Jan 2020

> Reviewer #1: The authors have addressed my previous comments. My suggestion for the Results sections would merely be to try to be more concise. All Results sections are very wordy and could easily be shortened by at least 30% simply by reducing words (example: "differed" instead of "differed greatly").

I have discussed with my co-authors and we believe it is important to keep the result sections as detailed as they are now. With so many analyses, the risk if we shorten is to loose clarity or precision (in our view “differed” and “differed greatly” is not the same).

Minor:

> Exp. D: title still says "strips" instead of "stripes"; please proof-read this section

We use the word strips throughout the manuscript, not only in the title of Exp. D. Our stimulus uses strips not stripes (small nuance). In fact, in the previous round the Reviewer pointed out that one time, and only once, I had used “stripes” instead of “strips”, which was corrected in the revised manuscript.

---

## [Editor Report · Decision Letter 2]

14 Jan 2020

Irrelevant auditory and tactile signals, but not visual signals, interact with the target onset and modulate saccade latencies

PONE-D-19-21342R2

Dear Dr. Vidal,

We are pleased to inform you that your manuscript has been judged scientifically suitable for publication and will be formally accepted for publication once it complies with all outstanding technical requirements.

With kind regards,

Nicholas V Swindale

Academic Editor

PLOS ONE

Additional Editor Comments (optional):

Re. 'strips' vs 'stripes' I tend to agree (if it is an issue) that 'strips' is the better term, given the stimulus. However the term 'stripes' is used in the Abstract and ought to be corrected (this should be done prior to production). I also tend to agree that the Results sections are OK as they are and it did not seem to me that they could be shortened much. I did find the use of the terms 'great' on lines 12 and 15 on page 8 a bit jarring - they could be replaced by 'considerable' or 'substantial' or 'a high degree of' - but it is a very minor point.
---

## [Editor Report · Acceptance letter]

4 Feb 2020

PONE-D-19-21342R2 

Irrelevant auditory and tactile signals, but not visual signals, interact with the target onset and modulate saccade latencies 

Dear Dr. Vidal:

I am pleased to inform you that your manuscript has been deemed suitable for publication in PLOS ONE. Congratulations! Your manuscript is now with our production department. 

With kind regards,

on behalf of

Dr. Nicholas V Swindale 

Academic Editor

PLOS ONE